# Lenvatinib activates anti-tumor immunity by suppressing immunoinhibitory infiltrates in the tumor microenvironment of advanced hepatocellular carcinoma

Masami Yamauchi [1✉], Atsushi Ono [1], Kei Amioka[1], Yasutoshi Fujii[1], Hikaru Nakahara[1,2], Yuji Teraoka[1], Shinsuke Uchikawa[1], Hatsue Fujino [1], Takashi Nakahara[1], Eisuke Murakami[1], Wataru Okamoto[1], Daiki Miki[1], Tomokazu Kawaoka[1], Masataka Tsuge[1], Michio Imamura[1], C. Nelson Hayes[1], Waka Ohishi[3], Takeshi Kishi[3], Mizuki Kimura[4], Natsumi Suzuki[4], Koji Arihiro[5], Hiroshi Aikata[6], Kazuaki Chayama [7,8,9✉] & Shiro Oka [1✉]

## Abstract

**Background** Lenvatinib, a multiple receptor tyrosine kinase inhibitor, might exert antitumor effects via tumor immune modulation. However, changes in the tumor immune micro-environment induced by lenvatinib are poorly understood. We investigated the effect of lenvatinib on immune features in clinical samples from patients with hepatocellular carcinoma.

**Methods** Fifty-one patients with advanced hepatocellular carcinoma who received lenvatinib monotherapy as first-line treatment were enrolled. We collected blood sample ($n = 51$) and tumor tissue ($n$, baseline/four weeks after treatment initiation/post-progression = 50/8/12). DNA, RNA, and proteins extracted from the tissues were subjected to multi-omics analysis, and patients were classified into two groups according to baseline immune status. Each group was investigated in terms of the dynamics of tumor signaling. We also longitudinally analyzed circulating immune proteins and chemokines in peripheral blood.

**Results** Here we show that lenvatinib has similar anti-tumor efficacy with objective response rate and progression-free survival in both Immune-Hot and Immune-Cold subtypes. Immune signatures associated with T-cell functions and interferon responses are enriched in the early phase of treatment, while signatures associated with immunoinhibitory cells are down-regulated along with efficient vascular endothelial growth factor receptor and fibroblast growth factor receptor blockades. These findings are supported by imaging mass cytometry, T-cell receptor repertoire analysis and kinetics of circulating proteins. We also identify interleukin-8 and angiopoietin-2 as possible targets of intervention to overcome resistance to existing immunotherapies.

**Conclusions** Our findings show the ability of lenvatinib to modulate tumor immunity in clinical samples of hepatocellular carcinoma.

### Plain language summary

Two types of therapy for liver cancer are immunotherapy and anti-angiogenic therapy. Immunotherapy helps the patient's immune system to attack the tumor. Anti-angiogenic therapy blocks the formation of new blood vessels (angiogenesis) in the tumor, and this type of therapy might also impact the immune system. We analyzed changes in the immune characteristics of human liver cancer samples induced by lenvatinib, an anti-angiogenic therapy. Patient outcomes on lenvatinib did not depend on the immune features of the tumor before treatment. However, immune characteristics of the tumors did change after treatment, and this may mean these tumors become easier to treat with immunotherapies. These findings help us to understand the effects of lenvatinib in liver cancer and whether, for example, it might be useful to combine this drug with immunotherapy.

---

A full list of author affiliations appears at the end of the paper.

Liver cancer has the seventh highest cancer incidence and represents the third leading cause of cancer death worldwide[1]. Most patients present with unresectable disease that has already spread at the time of diagnosis and shows poor prognosis[2]. Optimizing systemic therapies against hepatocellular carcinoma (HCC), which accounts for more than 90% of primary liver cancers, is problematic because of the scarcity of known genomic alterations[3] and a background of patient vulnerability mainly due to disturbed hepatic reserve. The recently approved programmed death 1 (PD-1) inhibition therapy has clearly improved survival outcomes for HCC with a tolerable safety profile when combined with the anti-vascular endothelial growth factor (VEGF) antibody bevacizumab[4].

Although the addition of bevacizumab represents a breakthrough in overcoming VEGF-mediated immunosuppression in the tumor microenvironment (TME)[5], further post-hoc analysis has suggested a lack of efficacy for subsets including HCC arising from non-alcoholic steatohepatitis[6] or HCC harboring Wnt/beta-catenin mutations[7,8]. In both situations, aberrant immune cell exclusion and excessive negative regulation of immune surveillance are regarded as the main causes of intrinsic resistance to immunotherapy. Development of combination therapies with synergistic agents that can alter the intricate biology of the tumor to be more responsive to immune checkpoint modification has been increasingly pursued.

Lenvatinib is an approved monotherapy for first-line HCC treatment and acts by inhibiting VEGF receptors (VEGFRs) 1–3, fibroblast growth factor (FGF) receptors (FGFRs) 1–4, platelet-derived growth factor receptor alpha, rearranged during transfection (RET) proto-oncogene, and the proto-oncogene c-KIT. Lenvatinib has also been expected to serve as a unique immunomodulator based on its marked potential for blockade of angiogenesis and its inhibitory activity against multikinases, including FGFR. In fact, the combination of lenvatinib and the anti-PD-1 inhibitor pembrolizumab considerably enhanced anti-tumor effects against HCC with improved objective response[9]. A broadly accepted but largely theoretical hypothesis is that anti-angiogenesis contributes to the remodeling of tumor immune systems not only by normalizing the abnormal tumor vasculature, but also by reactivating dysfunctional immune effector cells. However, evidence for this hypothesis has only been provided from preclinical studies[10,11] and the degree of immune modification according to the pre-existing immunity of the tumor remains poorly understood. The relevance of blockades beyond the VEGF pathway is also unknown.

To address these issues, we investigate phenotypic and functional changes in the TME by analyzing pre- and on-treatment tumor samples obtained from patients who received lenvatinib alone. We show the immunosupportive conversion of TME through a multi-omics approach uncovering genomic alterations within the tumor, immune transcriptome, T-cell receptor (TCR) repertoire, and immune proteome. We also demonstrate the value of longitudinal profiling of circulating immune proteins and chemokines, which enables us to monitor the state of TME less invasively.

## Methods

**Study overview.** This project comprised two clinical studies. The sample collection procedure was part of an interventional study registered under the University hospital Medical Information Network (UMIN) clinical trial registry system as UMIN000039887. Key inclusion and exclusion criteria for selecting candidates are listed in Supplementary Table 1. The Human Ethics Review Committee of Hiroshima University determined that taking tissue from liver tumor biopsies during the administration of therapeutic agents, especially in the context of routine medical care, was an invasive procedure for the participants. It stipulated that this clinical study be treated as an interventional study, but all drug administration was performed as standard treatment. The sample collection project from patients treated by lenvatinib as first-line therapy was completed in June 2020. The main outcomes including the relationship between histologically proven anti-tumor effect during treatment and survival outcomes are currently being analyzed for the publication. All analyses in this project were conducted retrospectively in accordance with the protocol of an observational study funded by Eisai Co., Ltd. registered as UMIN000044924. The sponsor provided financial support and collaborated with academic interpretation of the data and generation of the manuscript draft but did not participate in sample collection or data analysis. All patients provided written informed consent before enrollment. The Human Ethics Review Committees of Hiroshima University approved both studies in accordance with the Declaration of Helsinki. The approval numbers for the interventional and observational study were C2019-0291 and E2020-9250, respectively. Three patients, one with primary biliary cholangitis, one with non-viral chronic hepatitis, and one with microsatellite-high hepatocellular carcinoma, who received clinical practice at Hiroshima University Hospital, participated the preliminary analysis for TCR repertoire with written informed consent and the approval of The Human Ethics Review Committees of Hiroshima University.

**Patients and sample collection.** Fifty-one patients with unresectable HCC treated with lenvatinib between April 2018 and March 2020 at Hiroshima University Hospital met the eligibility criteria. Patients who had previously received any systemic treatment were excluded. Detailed patient characteristics are shown in Supplementary Table 2. All patients received a standard dose of lenvatinib monotherapy as the first-line treatment and continued until disease progression or the emergence of unacceptable toxicities. Clinical laboratory test results and occurrence of adverse events were continuously recorded and evaluated according to the National Cancer Institute Common Terminology Criteria for Adverse events (NCI-CTCAE) version 4.0. The best objective response and disease progression were determined by computed tomography in accordance with Response Evaluation Criteria in Solid Tumors version 1.1 (RECIST 1.1) and HCC-specific modified RECIST (mRECIST). In all patients, 5 ml of peripheral venous blood was collected at each timepoint. All patients received ultrasound-guided percutaneous liver tumor biopsies at baseline. Of those, eight patients underwent tumor biopsies at 4W, and 12 patients underwent biopsy at post-progression. To reduce the risk of hemorrhagic complications during lenvatinib administration, a 2-day drug holiday was held before and after each procedure.

**Tumor genotyping.** DNA of the primary tumor and adjacent non-cancerous liver was extracted from ten 5 μm-thick slides made from formalin-fixed paraffin-embedded (FFPE) samples acquired at baseline using the Maxwell RSC DNA FFPE Kit (Promega, Madison, US). The quantity and quality of FFPE-derived DNA samples were checked using a Qubit Fluorometer (Thermo Fisher Scientific, Waltham, US). DNA was fragmented and used for library construction according to the manufacturer's instructions. The 18 genes including *TERT promoter* (chr5, 1295151-1295313), *CTNNB1*, *TP53*, *ARID1A*, *ARID2*, *ALB*, *AXIN1*, *APOB*, *CDKN2A*, *RPS6KA3*, *FGFR1*, *FGFR2*, *FGF3*, *FGF4*, *FGF19*, *CCND1*, *ATM*, and *APC* were enriched using the Ion AmpliSeq Custom Panel (Thermo Fisher Scientific).

Sequencing was performed with paired-end reads on the Ion S5 XL system (Thermo Fisher Scientific). Sequencing reads were aligned to the hg19/GRCh37 reference sequence and analyzed using Ion Reporter Software (Torrent Suite Software; Thermo Fisher Scientific). Called variants were considered germline mutations if called in the control non-cancerous tissue or found in the dbSNP[12] database and 1000 Genomes Project[13].

**Immune transcript profiling in the TME.** RNA of the primary tumor was extracted from ten 5 μm-thick slides that were made from FFPE samples acquired at each timepoint using the Maxwell RSC RNA FFPE Kit (Promega). The quantity and quality of FFPE-derived RNA samples were checked using a Qubit 3.0 Fluorometer (Thermo Fisher Scientific) and Agilent 2100 Bioanalyzer (Agilent Technologies, Santa Clara, US). Purified RNA was hybridized with probes using the nCounter PanCancer Immune Profiling Panel, which is partially customized and enriches 780 genes, including surface markers of different immune cell types, common checkpoint inhibitors, cancer/testis antigens, and genes covering both adaptive and innate immune responses. Subsequently, digital counts and analyses were performed using the nCounter Digital Analyzer (NanoString Technologies, Seattle, US). Annotation and processing of data were performed using the GenePattern online platform[14], Gene Set Enrichment Analysis software[15], and a method for quantifying cell fractions from bulk tissue gene expression profiles (CIBER-SORT; Stanford University, US)[16].

**Imaging mass cytometry.** Three-micrometer-thick sections were made from FFPE specimens and mounted on slides for staining. Slides were rehydrated, and antigen retrieval was performed using a heat-mediated method (pH 9.0). A custom panel of antibodies for alpha-smooth muscle actin (SMA), beta-catenin, E-cadherin, granzyme B, Ki-67, FOXP3, CA9, CD3, CD4, CD8a, CD11c, CD14, CD20, CD68, CD163, VEGFR1, VEGFR2, FGFR2, PD-1, PD-L1, HLA-DR, and DNA intercalator was used to stain sections simultaneously. A list of corresponding clones and conjugated metal reporters used in the imaging mass cytometry experiments is shown in Supplementary Table 3. These sections were laser-ablated, and the acquired data were processed using the Hyperion Imaging System (Fluidigm, South San Francisco, US). Images were assessed and analyzed on MCD Viewer version 1.0.560.6 (Fluidigm). Enumerating antibody-positive cells in each region of interest (ROI) was performed automatically using binary images processed by ImageJ software version 2.3.0/1.53t[17].

**TCR repertoire analysis.** Freshly biopsied specimens were immediately frozen at −80 °C. Thirty samples obtained at baseline and on-treatment from 15 patients were analyzed. Total RNA was extracted using RNAlater-ICE Frozen Tissue Transition Solution (Thermo Fisher Scientific) according to the instructions provided by the manufacturer. After the PCR step was performed using a primer set that enriched the V, J, and C regions of *hTRA* (*human TCR alpha*), the PCR products were measured using a Qubit Fluorometer (Thermo Fisher Scientific) and checked by electrophoresis for quality. Eleven of 30 samples were below the required level for analysis and excluded from further analysis. Purified products were sequenced by Mi-Seq (Illumina, San Diego, US), and sequence data were analyzed by Repertoire Genesis version 20180912 software (Repertoire Genesis, Osaka, Japan). According to the number of sequenced unique reads for complementarity-determining region three (CDR3) of *TRA*, the diversity (Shannon-Weaver index H, Inverse Simpson's index 1/gamma, Pielou's evenness) or clonality (diversity evenness score [DE50]) of the TCR repertoire in each sample were calculated[18].

**Analysis of circulating cytokines and chemokines.** Separated serum was stored at −80 °C until batch analysis. Serum concentrations of analytes were evaluated using the Human Luminex Assay kit (R&D Systems, Minneapolis, US) according to the instructions from the manufacturer. This bead-based assay allows multiplex quantification of protein in samples. Using two commercially available panels (R&D Systems), serum levels of 25 proteins, including IFN-beta, IFN-gamma, TNF-alpha, granzyme B, IL-4, IL-6, IL-8, IL-10, IL-17, IL-18, CXCL9, CXCL10, CXCL11, FGF1, FGF2, FGF13, FGF-23, VEGF, angiopoietin-2, VEGFR1, VEGFR2, HGF, GM-CSF, CD163, and PD-L1 were measured.

**Statics and reproducibility.** Differences in mRNA expression levels were assessed using the Mann–Whitney $U$ test. Differences in levels of serum proteins were assessed using the Mann–Whitney $U$ test and Student's t-test. Differences in proportions of categorical variables were tested using Fisher's exact test. Correlations between analytes were assessed using Spearman's rank correlation. Sample clustering according to pathway analysis was performed using an unsupervised machine learning method, hybrid hierarchical k-means clustering[19]. For validation of the clustering method based on the transcriptome, RNA sequencing data from the liver HCC cohort from The Cancer Genome Atlas (TCGA-LIHC) were downloaded from the National Cancer Institute-Genomic Data Commons Data Portal[20]. Classifications of downloaded RNA-seq data to established HCC subtypes were performed using the Nearest Template Prediction platform[21], and were used for confirmation. PFS was estimated using the Kaplan–Meier method, and differences among subgroups with respect to candidate biomarkers were evaluated using the log-rank test. Uni- and multivariate Cox regression analyses were performed to predict outcomes based on potential biomarkers. Data integration across different omics sources was performed using the mixOmics platform[22,23]. All comparisons were considered significant at $p < 0.05$. All statistical analyses and graphical drawings were performed using R version 4.3.1 software with the latest package updates[24].

**Reporting summary.** Further information on research design is available in the Nature Portfolio Reporting Summary linked to this article.

## Results

**DNA and RNA analyses identify two TME subtypes of HCC.** This study was conducted using peripheral blood samples and biopsied liver tumor specimens from 51 patients with advanced, unresectable HCC treated using lenvatinib without PD-1 blockade. We set the following three timepoints for the analysis: baseline (before initiation of lenvatinib); at 4 weeks (4 W); and post-progression (Fig. 1a). Sample availability at each timepoint for the following analyses are shown in detail in Supplementary Fig. 1a. Tissue samples were available at baseline in all but one case ($n = 50$), while tissue samples during treatment were available for paired analysis at baseline and 4 weeks in 8 cases, and at baseline and post-progression in 12 cases. Blood samples ($n = 51$) were all comparable with no missing data at any time point.

We first analyzed target sequencing data of the tumor and surrounding non-tumoral tissue at baseline to reveal common mutations associated with HCC and to assess the characteristics of this population. Because the sequencing panel selected genes that are frequently affected in HCC, we found high-confidence somatic alterations despite the small amount of purified DNA. Consequently, we reproduced a common mutational spectrum of HCC with dominant *TP53* and *CTNNB1* mutations (Fig. 1b). No

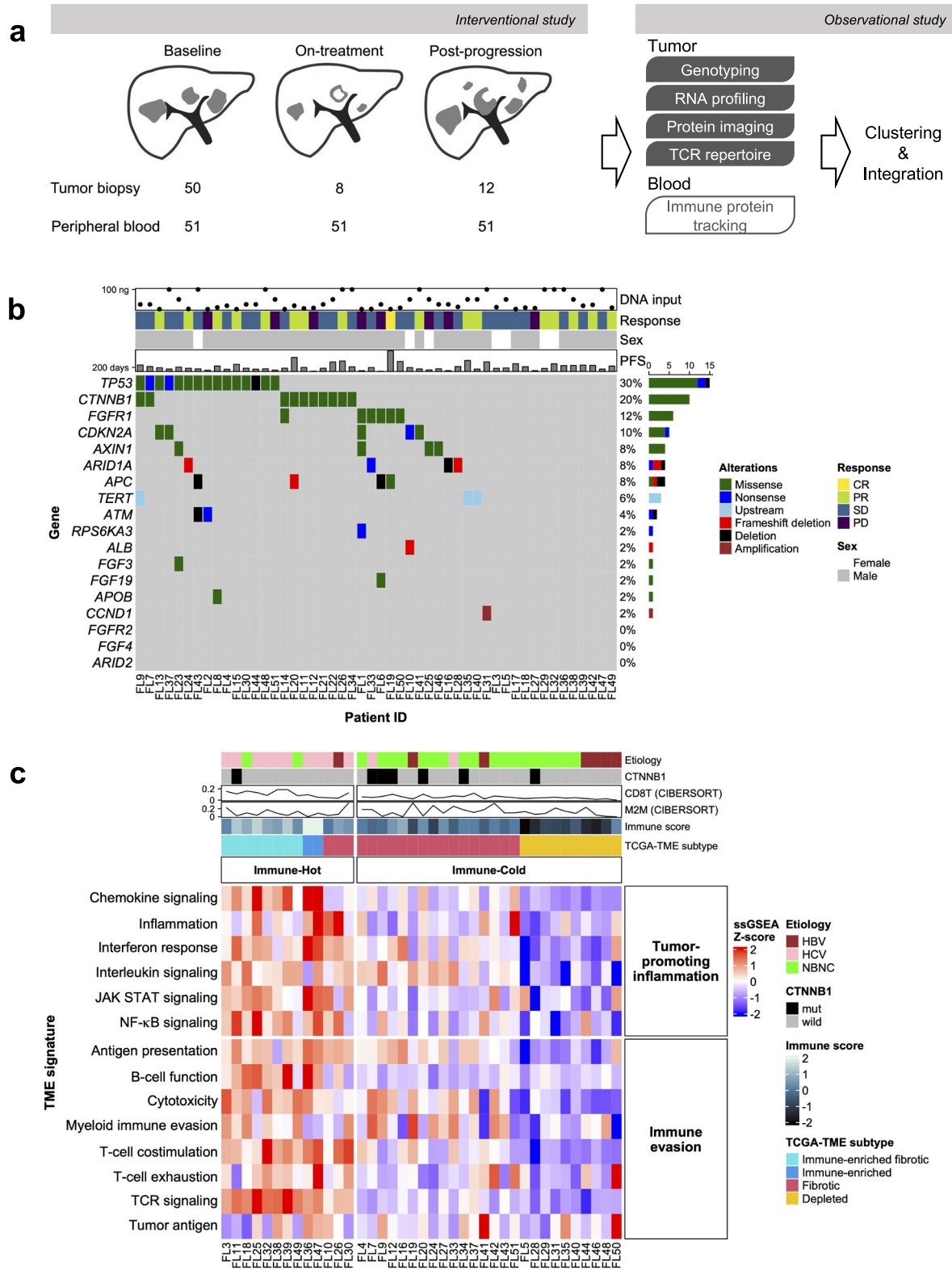

genes related to benefit in progression-free survival (PFS) were found.

Second, we classified specimens at baseline according to immune-related gene signatures. Digital quantitative nCounter values of 780 RNAs in each sample were submitted for single-sample Gene Set Enrichment Analysis (ssGSEA) using the nCounter Tumor Signaling 360 panel (Supplementary Data 1).

According to ssGSEA scores calculated through this pathway analysis, hybrid hierarchical k-means clustering (Supplementary Fig. 1b) recognized two immune subtypes, Immune-Hot ($n = 13$) and Immune-Cold ($n = 26$), characterized by distinct expression intensities in terms of immune evasion and tumor-promoting inflammation (Fig. 1c). A similar pathway analysis using well-defined gene signatures (Supplementary Data 2) identified four

**Fig. 1 TME immune subtypes at baseline with distinct proliferative features. a** Scheme of the study. **b** Mutation analysis. Annotation columns over the oncoplot show the amount of purified DNA processed to targeted sequencing, objective response to subsequent lenvatinib therapy confirmed by RECIST 1.1, sex, and progression-free survival. **c** Transcriptome clustering. Two immune subtypes of the TME, Immune-Hot and Immune-Cold, are shown in the clustered heatmap. Annotation columns show the etiologies of background liver disorders, the mutation status of beta-catenin (*CTNNB1*), CIBERSORT estimation scores for the fraction of infiltrated CD8+ T lymphocytes and M2-type macrophages, Immune score (Yoshihara K, 2013), and TCGA-TME subtype (Bagaev A, 2021). TCR T-cell receptor, PFS progression-free survival, RECIST Response Evaluation Criteria in Solid Tumors, CR complete response, PR partial response, SD stable disease, PD progressive disease, CD8T CD8+ T lymphocytes, M2M M2-type macrophages, TCGA The Cancer Genome Atlas, TME tumor microenvironment, HBV hepatitis B virus, HCV hepatitis C virus, NBNC non-B, non-C hepatic disorder including nonalcoholic fatty liver disease; mut mutant.

TCGA-TME subtypes (immune-enriched fibrotic, immune-enriched, fibrotic, and immune-depleted)[25], which were closely related to our immune subtype classification (upper column, Fig. 1c). *CTNNB1* mutations in tumors or non-viral etiologies of background liver injury appeared to be associated with the Immune-Cold microenvironment, as described in previous reports[6,7]. Another established score for quantifying immune cells in the TME, Immune score[26], also showed a strong correlation with our clustering results (upper column, Fig. 1c). To further validate the reliability of our panel-based assay, which had limited mounted RNA, we cross-referenced a large RNA-seq dataset from the TCGA-LIHC cohort using the same approach (Supplementary Fig. 1c). The classifications made by the nCounter RNA set were generally consistent with the results obtained using the TCGA-TME RNA set except for a small population within the TCGA-TME-Fibrotic subtype, where the signatures of anti-tumor immune infiltrates were relatively overexpressed. Furthermore, the Immune-Hot subtype predominantly included cases classified as Immune-specific class by Sia[27] and HCC-subtype-S1 by Hoshida[28]. In contrast to the Immune-Hot subtype, the Immune-Cold subtype appeared to correlate with Hoshida-subtype-S2, which is a more aggressive subtype characterized by activation of proliferation pathways.

**Lenvatinib shows similar anti-tumor efficacy regardless of TME immune subtype before treatment**. To evaluate the characteristics of each subtype, we compared the expression levels of tumor signaling and tumor microenvironment-related signatures at baseline in Immune-Hot and Immune-Cold (Fig. 2a). As a result, the expression levels of several signatures were significantly different between each subtype. In the assessment of tumor signaling-related gene signatures at baseline, both immune subtypes displayed distinctive characteristics regarding tumor proliferation. The Immune-Cold subtype was associated with increased tumor growth activities. On the other hand, the enrichment of angiogenesis and hypoxia-induced signaling did not differ between subtypes. Other signatures regarding tumor growth or invasion were generally activated in the Immune-Cold subtype (Supplementary Fig. 2a).

Next, we examined the clinical survival benefit of lenvatinib in each subtype. Despite considerable differences in tumor biology, lenvatinib yielded similar disease control rates and survival benefits for both subtypes (Fig. 2b, c). Because previous reports have noted the lack of options for immunotherapy in the Immune-Cold TME, effectiveness as a targeted agent against the Immune-Cold subtype can be regarded as an important advantage of lenvatinib.

We also checked the phenotype of two immune subtypes in terms of protein-level expression. As shown in Fig. 2d and Supplementary Fig. 2b, c, the Immune-Cold subtype has low-level expression of immune-related proteins in the TME compared to the Immune-Hot subtype. Of note, some cases have modest to high levels of activation of immune infiltrates, especially in adjacent non-tumor liver tissue and stroma accompanied by activated VEGF and FGF signaling.

**Lenvatinib activates innate and acquired anti-tumor immunity in TME**. To shed light on the key immunologic phenomena that underlie the complex TME system, we focused on eight cases (Fig. 3a) with paired pre- and on-treatment (4W) tumor biopsy samples and explored early changes in the immune-related transcriptome. In all eight cases, objective response to lenvatinib was confirmed at least briefly as a partial response by computed tomography in the subsequent observation periods.

We first identified differentially expressed genes in the nCounter RNA dataset between baseline and on-treatment samples (Fig. 3b). Reflecting the response to cellular damage caused by systemic chemotherapy, upregulation was observed for *MYD88* and *OAS1*, which are involved in activation of innate immunity pathways associated with damage-associated molecular patterns (DAMPs)[29] or stimulator of interferon genes (STING)[30]. Further, upregulation of *LTBR*, which is crucial for increasing the infiltration or migration of T lymphocytes in tumor sites[31], was also observed. On the other hand, *STAT3*, a transcription factor that suppresses the anti-tumor immune response[32] showed compensatory activation, presumably in response to stimulation of antitumor immunity in the TME. Typical genes associated with angiogenesis and VEGF-mediated immunosuppression, including *FLT1* (which encodes VEGFR1), *CD34*, and *ANGPT2* were significantly decreased, reflecting VEGFR blockade by lenvatinib. Decreased expression was also observed for *IL3RA* (*CD123*), a marker for plasmacytoid dendritic cells as a known immunosuppressive subgroup of dendritic cells[33]. Interestingly, expression of *FZD4*, which generally increases after anti-VEGF monotherapy and promotes resistance through the activation of FGF signaling[34], was clearly suppressed, perhaps due to simultaneous FGF-FGFR pathway blockade by lenvatinib.

From the perspective of the network transcriptome, gene set enrichment analysis more clearly and concretely revealed the modulation of antitumor immune function in the TME (Fig. 3c, d). Significant upregulation of immune response-related signatures such as antigen presentation, myeloid cell evasion, inflammation, and interferon response were observed in on-treatment samples. Upregulation of the T-cell signature with exhaustion markers suggests the re-invigoration of resident T cells that have already infiltrated the locoregional TME at the start of systemic therapy. In contrast, signatures associated with tumor signaling, including cell cycle, WNT signaling, and FGFR signaling, and the TME, including cell adhesion, extracellular matrix signaling, and VEGF signaling, were downregulated in accordance with the single gene analyses mentioned above (Supplementary Fig. 3a). These results could concordantly support the hypothesis that lenvatinib activates the exhausted anti-immune condition of the TME through the inhibition of VEGFR- and FGFR-mediated immunosuppression.

To capture changes in specific immune cell fractions in on-treatment samples, we performed further pathway analysis using the TCGA-TME functional gene expression signatures[25]. In both Immune-Hot (3 cases) and Immune-Cold subtypes (5 cases), increases in the signatures of anti-tumor infiltrated cells and anti-tumor-related functions (antitumor cytokines, co-activation

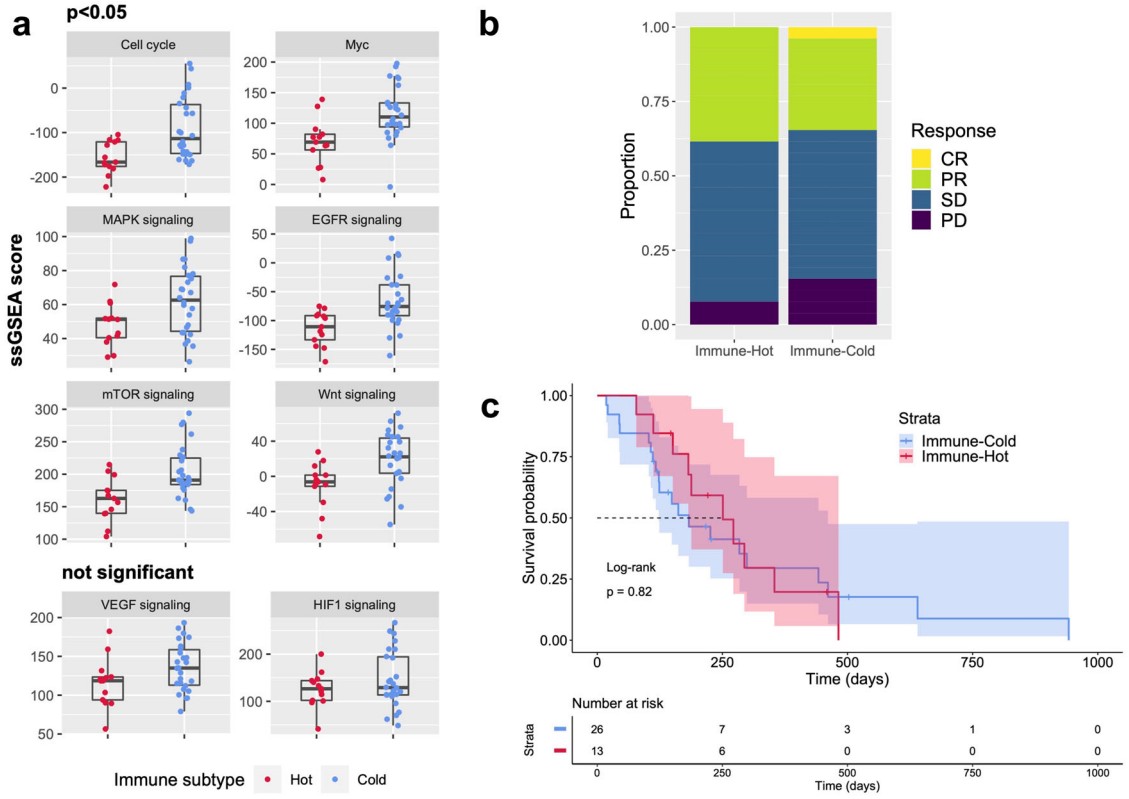

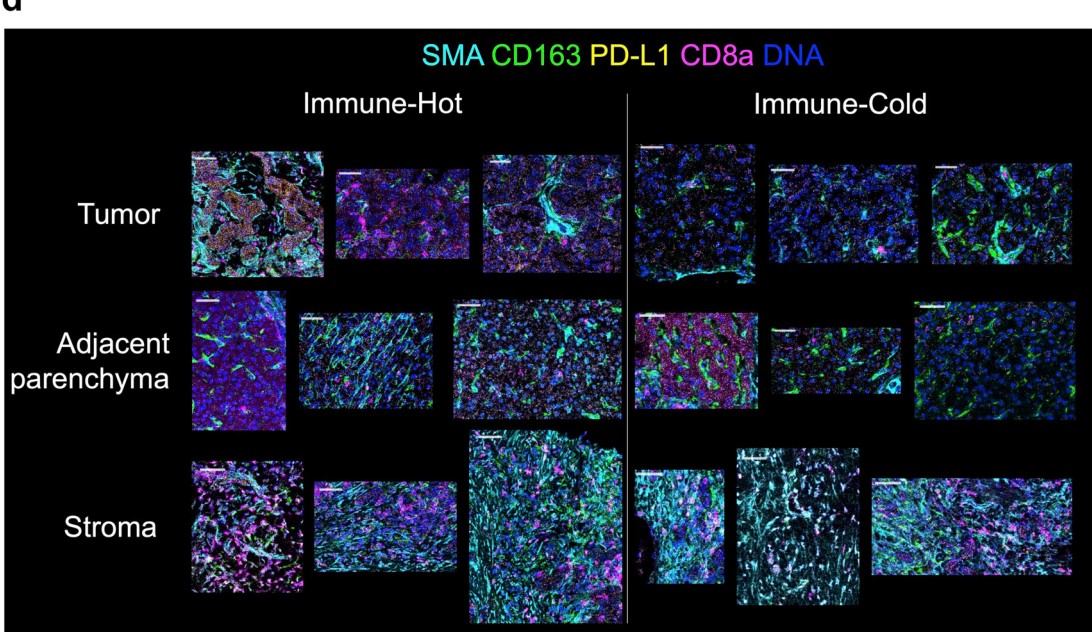

**Fig. 2 RNA and protein-level characteristics and survival benefit of lenvatinib in both subtypes. a** Box plots showing the characteristics of each immune subtype regarding tumor signaling and tumor microenvironment. Signatures including cell cycle, Myc MAPK signaling, EGFR signaling, mTOR signaling, and Wnt signaling differ significantly ($p < 0.05$) between subtypes. VEGF and HIF1 signaling show comparable levels of expression among subtypes ($p = 0.7$ and 0.5, respectively). Each dot represents a single case. Boxes show the median (horizontal line), and the 25th and 75th percentiles. Whiskers indicate the 75th percentile plus 1.5 × interquartile range and the 25th percentile less 1.5 × interquartile range. Dots beyond whiskers are outliers. **b** Objective response confirmed by RECIST 1.1. Overall response rates (CR + PR) were 38% for the Immune-Hot subtype and 35% for the Immune-Cold subtype ($p = 1.0$). **c** Kaplan–Meier curves for PFS with 95%CIs according to immune subtypes. Median PFS did not differ significantly between subtypes (183 days for Immune-Cold vs. 251 days for Immune-Hot subtype). The dotted line indicates the 50% probability of PFS. **d** Representative mass cytometry images of each immune subtype focusing on the immune status of the TME separating the tumor and adjacent non-tumor areas. Scale bars, 100 μm. $n = 13$ for Immune-Hot and $n = 26$ for Immune-Cold (**a–c**). ssGSEA single-sample Gene Set Enrichment Analysis, RECIST Response Evaluation Criteria in Solid Tumors, CR complete response, PR partial response, SD stable disease, PD progressive disease, PFS progression-free survival, 95%CI 95% confidence interval, TME tumor microenvironment.

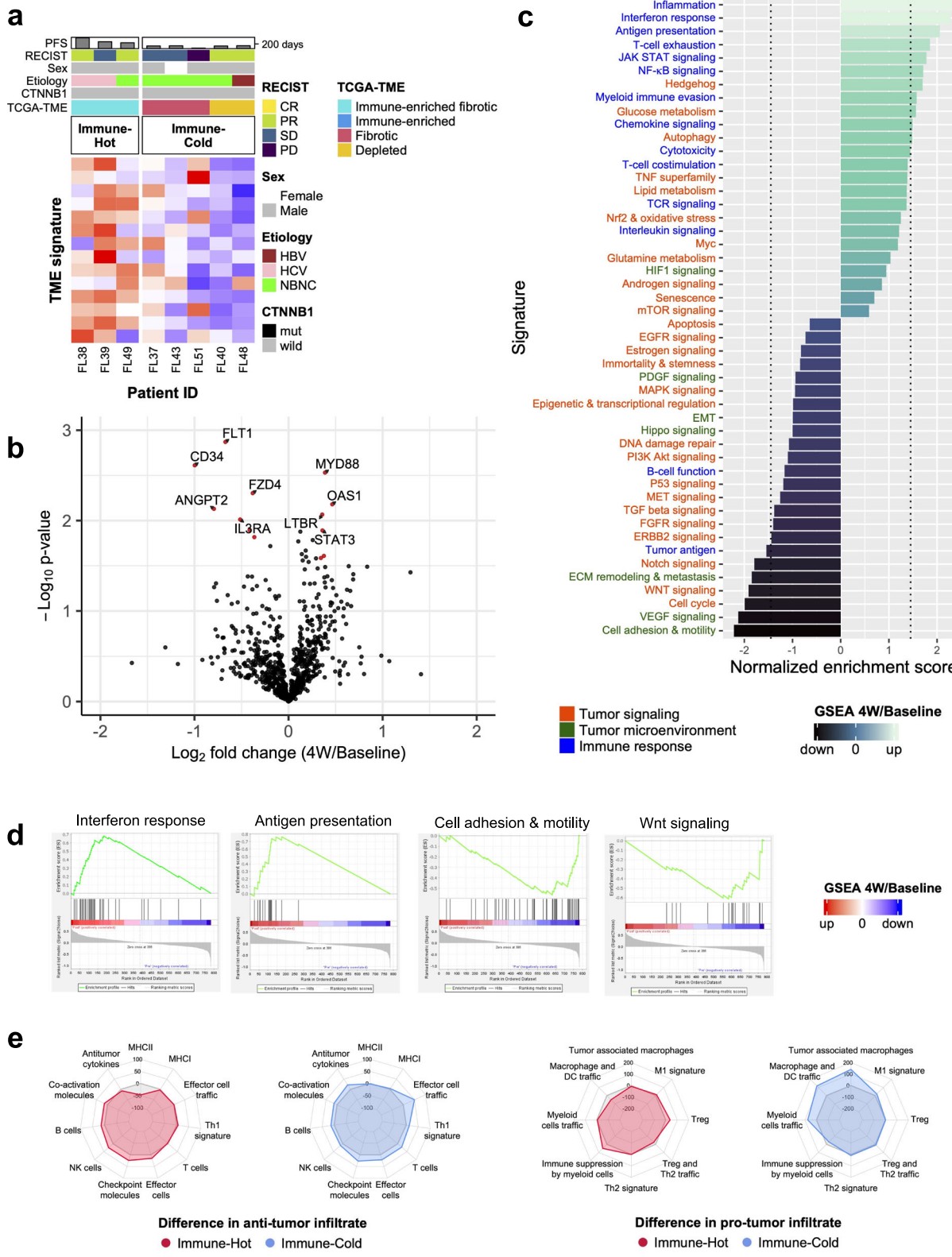

molecules, and effector cell traffic) at 4W were observed. The degree of these changes was greater in the Immune-Cold subtype (Fig. 3e). Compared with the pro-tumor infiltrate signature, marked increases were seen related to myeloid cell function (tumor-associated macrophages, macrophage and dendritic cell traffic, and myeloid cell traffic) in the Immune-Cold subtype. Because the scores of signatures derived from core inhibitory

immune cells (immune suppression by myeloid cells, regulatory T-cell (Treg) and type-2 T helper cell (Th2) signatures) did not change during treatment, the increase in the myeloid cell signature in the TME of the Immune-Cold subtype can be regarded as a result of anti-tumor responses like phagocytic interactions[35]. Analysis using the nCounter signature in the same manner showed corroborating results (Supplementary Fig. 3b).

**Fig. 3 Immunity-related transcriptomes are activated during lenvatinib therapy in the TME. a** Heatmap showing clinical characteristics of 8 cases with baseline and 4-week samples analyzed in pairs. Median age was 70.5 years old (47–84) and median PFS was 251 days (95% CI, 123-NA). **b** Volcano plots showing results for differential RNA expression as quantified by the nCounter system for eight paired (4 W/pre-treatment) samples. Red, $p < 0.05$ and over a $\log_2$ fold change cut-off value of 0.3 (23% increase) or −0.3 (23% decrease). **c** Visualization of normalized enrichment score for each signature with the nCounter Tumor Signaling Panel 360 (4W/pre-treatment). Dotted line indicates the threshold for the statistical significance of up- and downregulation in 4W samples ($P < 0.05$). **d** Representative enrichment plots. Activation of immune response (interferon response, antigen presentation), inactivation of tumor microenvironment (cell adhesion and motility) and inactivation of tumor signaling (Wnt signaling) in on-treatment samples. **e** Radar plots showing differences in scores of immune infiltrates (ssGSEA value at 4W minus that at baseline). $n = 8$ pairs (baseline and 4W), (**a–d**). PFS progression-free survival, RECIST Response Evaluation Criteria in Solid Tumors, CR complete response, PR partial response, SD stable disease, PD progressive disease, TCGA The Cancer Genome Atlas, TME tumor microenvironment, HBV hepatitis B virus, HCV hepatitis C virus, NBNC non-B, non-C hepatic disorder including nonalcoholic fatty liver disease, mut mutant, GSEA Gene Set Enrichment Analysis.

In addition, we performed a partial analysis on the post-progression samples. Of note, positive immunological responses were externalized only in the early phase of treatment (4W). In the post-progression refractory period, inactivation of anti-tumor immunity and resistance to the suppression of tumor signaling by lenvatinib might occur even though VEGF and FGFR signaling were persistently inhibited (Supplementary Fig. 4a, b).

**In-depth proteomics reveal reconstruction of immune cells in the TME.** Next, we performed protein-level analyses focusing on the dynamics of immune cells in tumor lesions by imaging mass cytometry. From each specimen, we selected 2–4 regions of interest (ROIs) with areas of 0.2–0.3 mm$^2$ and compared the absolute number of immune cells per unit area between the baseline and 4W samples. In line with former results from RNA analysis that suggested immune activation in the TME by lenvatinib, the number of CD3$^+$CD4$^+$ cells (CD4$^+$ T lymphocytes) tended to increase. However, the number of CD3$^+$CD4$^+$FOXP3$^+$ cells (Tregs) did not change much, and consequently the ratio of Tregs to CD4$^+$ T lymphocytes significantly decreased during treatment (Fig. 4a, d, Supplementary Fig. 3c). Similarly, the ratio of M2-type macrophages (CD68$^+$CD163$^+$ cells) to pan-macrophages (CD68$^+$ cells) clearly decreased (Fig. 4b, d, Supplementary Fig. 3c) during treatment. Probably in connection with this proportional suppression in immune-inhibitory cells, a significant increase was observed in the number of cytotoxic T cells (CD3$^+$CD8$^+$ cells) in the same samples (Fig. 4c, d, Supplementary Fig. 3c). Generalization across multiple cases could not be confirmed because of the limited number of ROIs; we also had an interesting observation suggesting suppression of beta-catenin signaling and activation of the TME (Fig. 4e).

TCR repertoire analysis allowed us to explore the diversity of T cells in the tumor. Although the number of paired samples available for analysis was limited, in two of three pairs, three diversity indexes, which are widely used for comparing diversity between samples, were increased at 4W (Fig. 4f). Furthermore, an increase in the clonality of the repertoire, a possible tumor-specific response, was also observed in on-treatment samples. Compared to controls, including two types of non-cancerous liver inflammation, repertoire diversity in on-treatment samples was outstanding. The degree of variation in the repertoire was comparable to that of microsatellite instability-high HCC (Supplementary Fig. 5b), which led to complete response on pembrolizumab treatment[36]. The marked heterogeneity shown in the repertoire at the very early phase of lenvatinib treatment suggests that the reconstruction of TME arose from resident immune cells, which already recognized tumor antigens and fell into exhaustion and dysfunctional states[37].

**Exploratory analysis of blood as less-invasive biomarkers for presuming TME and predicting prognosis.** In the context of optimization in systemic therapy for HCC, we returned to whole-patient analysis and explored serum assays for circulating chemokine/angiogenic factors. Of 25 serum proteins quantified by the Luminex system, the CXCL10 level at baseline allowed the differentiation of two immune subtypes (Fig. 5a, Supplementary Fig. 6a). Although sensitivity analysis showed that the statistics might be affected by possible outliers, the validity of serum CXCL10 as a biomarker is worth validation in a larger number of cases or in combination with other chemokines for the following reasons. As a key modulator for chemoattraction of immune cells, promotion of T-cell function, and antitumor activity in the local TME[38], CXCL10 may offer a surrogate for presuming the tumor immune subtype.

We then explored these serum proteins at baseline as less-invasive biomarkers for predicting prognosis. Serum concentrations of FGFs, VEGF, and soluble VEGFRs, which can be directly affected by the pathway inhibition afforded by lenvatinib, showed no associations with PFS. Notably, high levels of serum interleukin-8 (IL-8) and angiopoietin-2 were identified as strong negative prognostic indicators for the entire cohort (Fig. 5b, Supplementary Fig. 6b). Although levels of IL-8 and angiopoietin-2 displayed only weak Spearman correlations (Supplementary Fig. 6c, d), deterioration in PFS after stratification by these two proteins remained independent for both immune subtypes (Supplementary Fig. 6e). High concentrations of IL-8[39,40] and angiopoietin-2[41] in blood have recently been considered as potential signals for the decreased efficacy of immunotherapy.

**Comprehensive analysis of serum circulating proteins validates immunologic transformation in the TME.** We confirmed immune modifications during treatment using serum pairs at baseline and 4W. When focusing on the fluctuations in the level of 25 serum proteins with attention to immunity, the level of interferon gamma, which is crucial for exploiting the antitumor effects of immunotherapy[42], was significantly increased under lenvatinib monotherapy (Fig. 5c). Similarly, significant increases were seen in levels of IL-6, as a key player in the activation, proliferation, and survival of lymphocytes during active immune responses in the TME[43]. Elevations of CXCL9 and CXCL10, both of which facilitate the attraction of effector cells to the TME, match the context of immune activation by lenvatinib. Levels of granzyme-B, an indispensable factor in T-cell-mediated tumor killing[44], also increased. Although studies performing serial measurements of serum concentrations of granzyme-B during treatment in clinical settings remain quite rare, levels of granzyme B in peripheral blood may offer a surrogate for T-cell function in the TME. In the present study, only levels of IFN-beta decreased among potential activators[45] of the TME.

On the other hand, levels of the immunoinhibitory circulating chemokine/angiogenic factors IL-8 and angiopoietin-2 decreased significantly during treatment. Decreases in both IL-8 and angiopoietin-2 can be explained by the functional suppression of IL-8-producing immune inhibitory cells and antiopoietin-2-producing vascular endothelial cells, which have been demonstrated

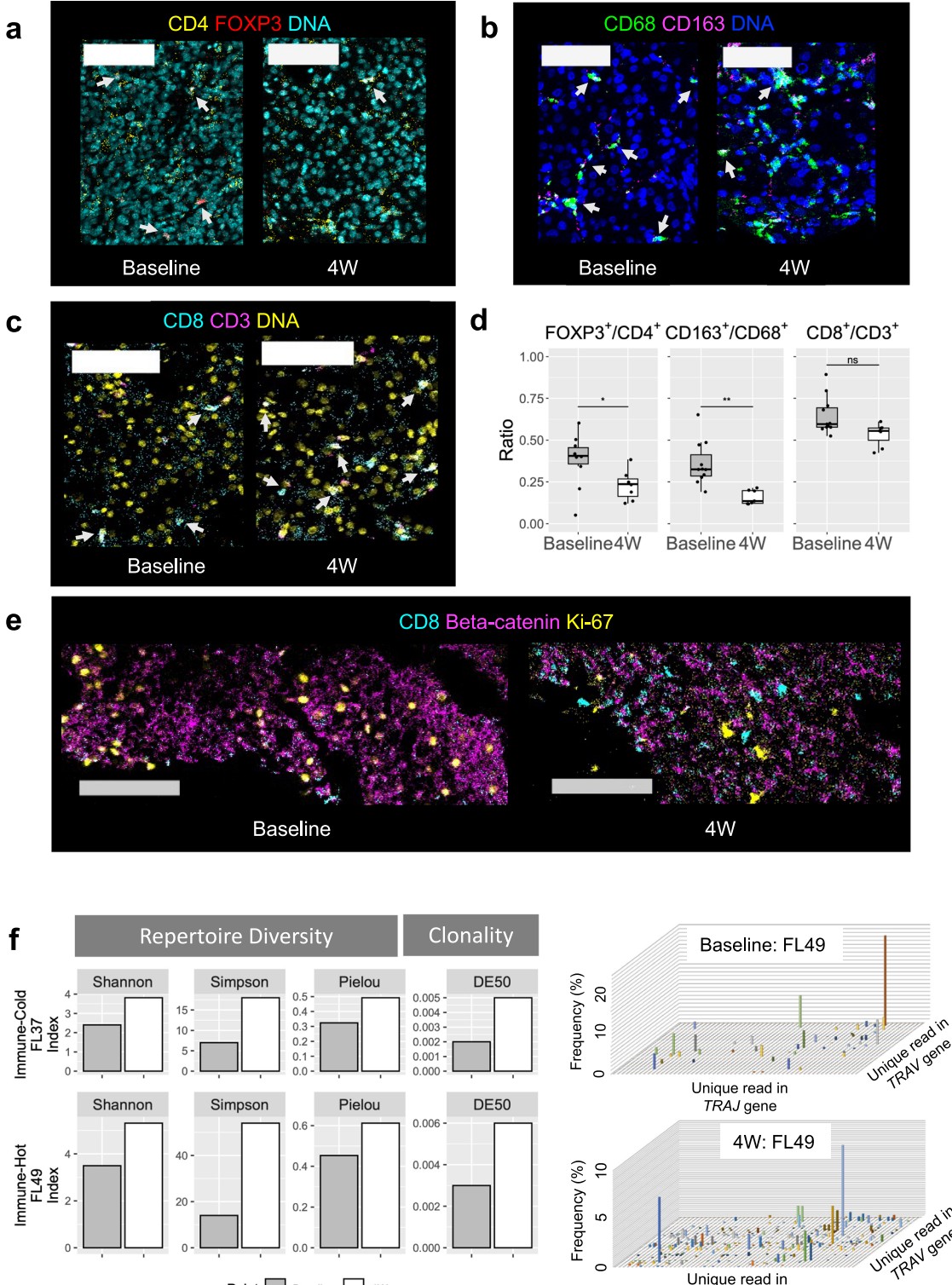

to be immunomodulatory effects of lenvatinib. The degree of suppression was greater in the baseline IL-8-high and angiopoietin-2-high subgroups (Fig. 5d). The trends of increase and decrease in circulating proteins are similarly observed in both the Responder and Non-Responder groups (Supplementary Fig. 7a, b).

**Multi-omics integration facilitates deeper understanding of the TME.** We confirmed the relationship between individual single-omics analyses through several data coupling approaches.

Correlation analysis comparing the TCR repertoire and the TME transcriptome at baseline showed a negative relationship between diversity indexes (Shannon, Simpson and Pielou) of repertoire and immune evasion signatures by myeloid cells (Fig. 6a). It also showed a negative correlation between repertoire clonality (DE50) and the T-cell exhaustion signature including *PD-1*, *LAG3*, and *EOMES* genes. The large number of paired samples when comparing immuno-transcriptome and circulating proteins in blood allowed for data integration based on multivariate

**Fig. 4 Proteomics reveal suppression of immuno-inhibitory cells and reinvigoration of T-cell function. a** Representative mass cytometry images of HCC tumor at baseline and on-treatment, showing overlay of CD4 (yellow), FOXP3 (red) and DNA (cyan). White arrows show CD4 and FOXP3 double-positive cells (Treg) among CD4 single-positive lymphocytes. Scale bars, 100 µm. **b** Representative images showing overlay of CD68 (green), CD163 (magenta) and DNA (blue). White arrows show CD68 and CD163 double-positive cells (M2-type macrophage) among CD68-positive macrophages. Scale bars, 100 µm. **c** Representative images showing overlay of CD8 (cyan), CD3 (magenta) and DNA (yellow). White arrows show CD8 and CD3 double-positive cells (cytotoxic T-cell) among CD3-positive T lymphocyte. Scale bars, 100 µm. **d** The changes in proportion of CD4- and FOXP3-positive cells to CD4-positive cells, CD68- and CD163-positive cells to CD68-positive cells, and CD8- and CD3-positive cells to CD3-positive cells during treatment. Boxes show the median (horizontal line), and the 25th and 75th percentiles. Whiskers indicate the 75th percentile plus 1.5×interquartile range and the 25th percentile less 1.5×interquartile range. Dots beyond whiskers are outliers. Absolute counts of each cell in the ROIs are displayed in Supplementary Fig. 3c. $n = 10$ for baseline and $n = 7$ for 4W (FOXP3+/CD4+). $n = 11$ for baseline and $n = 7$ for 4 W (CD163+/ CD68+, CD8+/CD3+). **e** Representative images showing decreased beta-catenin expression and increased CD8-positive cells during treatment overserved in tumor harboring *CTNNB1* mutation (FL12). Scale bars, 100 µm. **f** Bar plots showing changes in diversity and clonality scores for *T-cell receptor alpha (TRA)* repertoires during treatment. Each bar represents a single value calculated from the repertoire sequencing data for each case (FL37 and FL49). The right two panels show representative three-dimensional plot of *TRA* repertoires in Immune-Hot case (FL49), showing increasing diversity of *TRA* during treatment. The x- and y-axes indicate the sequenced positions of TRAJ and TRAV regions and the z-axis indicates frequencies of each TRA combination. $^*p < 0.05$; $^{**}p < 0.01$, ns not significant, Shannon Shannon-Weaver index H′, Simpson Inv. Simpson's index 1/λ, Pielou Pielou's evenness, DE50 Diversity Evenness score.

regression[22,23]. The analysis extracted not only several expected factors (e.g., serum CXCL10 and Granzyme B) in discriminating the immune subtype, but also unexpected factors such as FGF23 (Fig. 6b, Supplementary Fig. 7c–e). Network analysis showed that serum FGF23, a ligand for FGFR4, may be closely related to serum CXCL10 via chemokine (*CCL5*) and T-cell-associated molecule (*TRAT1*) expression in the microenvironment (Fig. 6c).

## Discussion

Tracking functional dynamics of the tumor-immune ecosystem during systemic therapy of advanced cancer in an unbiased manner has long been challenging. This study revealed several changes in the different immune subtypes of HCC, while lenvatinib showed comparable antitumor effects in both subtypes. To the best of our knowledge, we provide the first evidence of positive immune modification induced by monotherapy with a molecularly targeted agent lenvatinib together with inhibition of angiogenesis in a clinical sample. Our integrative multi-omics approach and sequential observations in different immune subtypes provide putative mechanistic insights to better understand the nature of the TME and to develop new strategies to enhance the efficacy of immunotherapy.

The most important finding in this study was that lenvatinib monotherapy itself has the potential for immune editing, even when an immune-excluded status has already been established in the TME. In recent biomarker research from landmark clinical trials[4,46] assessing atezolizumab plus bevacizumab therapy for patients with HCC, partial evidence was obtained for the inhibition of VEGF-mediated Treg proliferation and inhibition of myeloid cell inflammation during combined immunotherapy[47]. In a phase 1b trial assessing neoadjuvant combined immunotherapy against HCC, the targeted drug cabozantinib (an inhibitor of VEGFR2, AXL Receptor Tyrosine Kinase [AXL], and hepatocyte growth factor receptor [c-MET]) was separately shown to contribute to T-cell activation in the TME through the suppression of chemokines[48]. We added new evidence of immuno-activation in the tumor resulting from VEGFR and FGFR blockade by lenvatinib. Our data from transcriptome and imaging mass cytometry successfully corroborated the findings from those studies by showing the functional and proportional suppression of Tregs and M2-type macrophages and reinvigoration of T cells in clinical HCC samples. This coordination is persuasively supported by the kinetics of immune-related circulating proteins in peripheral blood. Moreover, the present study not only seamlessly explains a series of immunomodulation events in the TME but also contributed a new perspective for perceiving the pre-treatment immunity of HCC. According to the

whole genome sequencing analysis using liver cancer samples, intrinsic immunosuppression, a major barrier to the enhancement of immunotherapy, is driven by the mutually exclusive mechanisms of tumor-associated macrophages, Tregs, *CTNNB1* mutation, and cytolytic activity[49]. Our observations suggest that at least the first two mechanisms have some degree of plasticity and may be regulated by lenvatinib regardless of the pretreatment immune subtype. To develop an optimized counterpart to anti PD-1 therapy, further elucidation of pretreatment immunity and visualization of the power of agents to intervene in pre-existing immunity is warranted.

It is also important to note the utility of circulating immune proteins when monitoring the TME. In clinical settings, the technical difficulty or invasiveness of tumor biopsies limits the ability to rely heavily on tissue-based information. Although CXCL9 and CXCL10 are broadly regarded as essential mediators that facilitate T-cell migration into tumors[50], the importance of the concentrations detected in peripheral blood remain unclear. It is reported that FGFR inhibition activates the IFN-gamma pathway in HCC cells and enhances secretion of chemokines, including CXCL10[51]. Our observations suggest that this positive immunomodulation of lenvatinib occurs in human tumor samples and, furthermore, may be detected from peripheral circulation. In addition, observed trends in circulating protein levels regardless of tumor response suggest that the immune modification induced by lenvatinib is not due to its antitumor effect, but rather is evoked by inhibition of signaling pathways. We, therefore, propose that serum CXCL10 concentrations may serve as a surrogate for tumor immune subtypes and that levels may fluctuate with serum CXCL9 concentrations, reflecting the extent of immune reconstitution by anti-angiogenic therapy. Of course, the validity of these translational findings should be determined in combination with larger cohorts, other solid tumor cases, or in combination with other serum analytes. Recent advances in late-stage systemic therapy and patient supportive care have led to gradual increases in overall survival among patients with HCC. As a result, the add-on effects of immune checkpoint inhibitors combined with targeted therapies have become difficult to simply detect in the clinical trial setting of first-line therapy[52]. Hence, proper patient stratification based on the estimated TME status through less-invasive repeatable assessment by blood examination would represent a major advantage in the design of optimal clinical trials.

We would like to highlight an unexpected but interesting feature of lenvatinib; namely, its potential to suppress the immunoinhibitory circulating chemokine/angiogenic factors IL-8 and angiopoietin-2. Although the decrease in IL-8 secreted from cancer cells by the antiangiogenic agent sunitinib was observed in

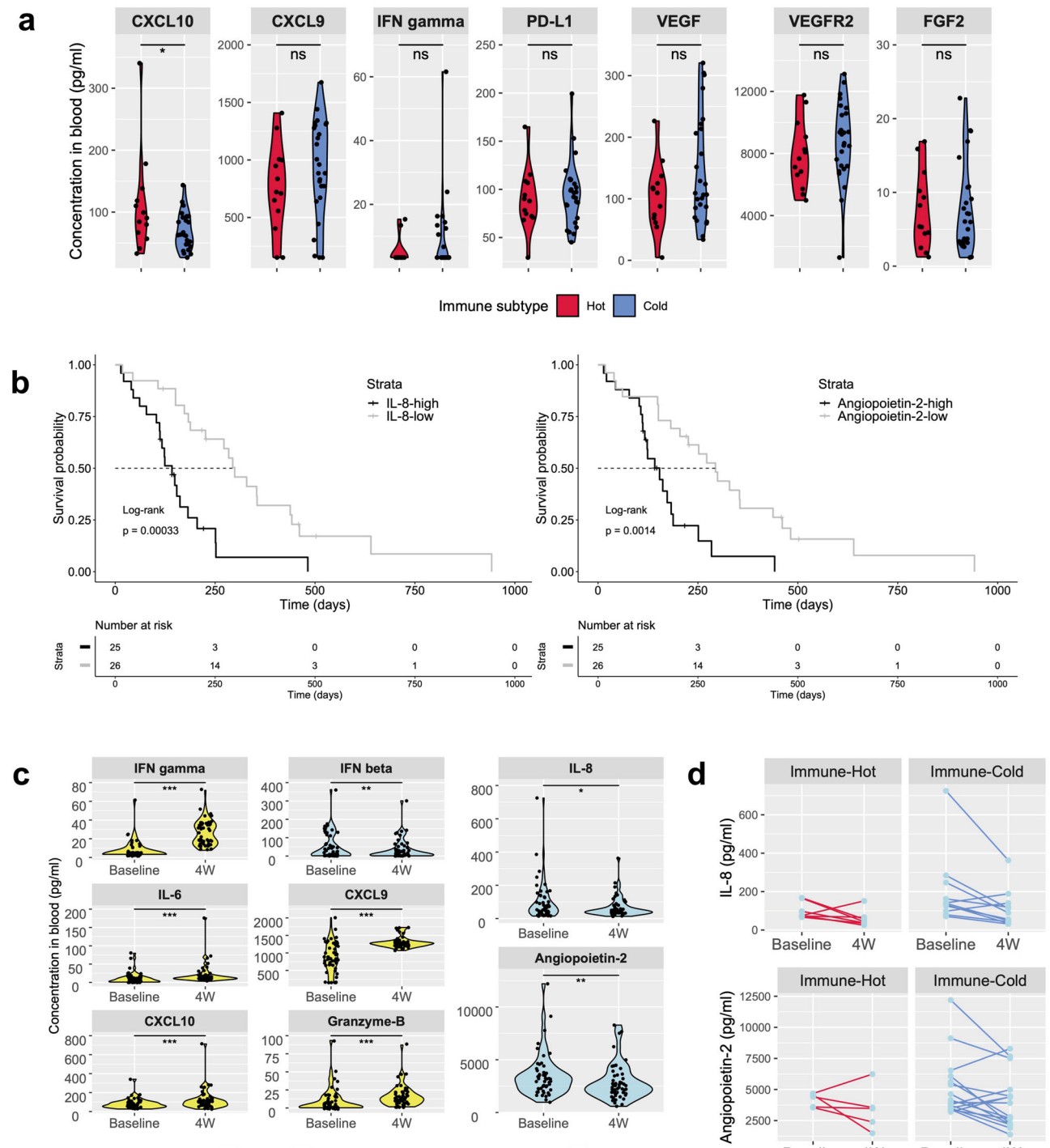

**Fig. 5 Relationship between the TME and immune-related circulating proteins in blood. a** Levels of circulating CXCL10 are significantly higher for the Immune-Hot subtype than for the Immune Cold subtype ($P = 0.022$). The result of sensitivity analysis is shown in Supplementary Table 4. Levels of other proteins are not significantly different between the two subtypes (continues to Supplementary Fig. 6a). Each dot represents a single case. $n = 13$ for Immune-Hot and $n = 26$ for Immune-Cold. **b** Kaplan–Meier curves for PFS stratified by baseline levels of serum IL-8 and angiopoietin-2. Median PFS for IL-8-high and IL-8-low subgroups are 142 days (95%CI, 112–205 days) and 299 days (95%CI, 226–442 days), respectively. Median PFS of angiopoietin-2-high and angiopoietin-2-low subgroups are 154 days (95%CI, 117–251 days) and 294 days (95%CI, 205–461 days), respectively. The dotted line indicates the 50% probability of PFS. $n = 51$. **c** Violin plots showing significant changes in serum concentrations of immune-related chemokines and cytokines detected by the Luminex system. P-values for each analyte are less than 0.01, except for IL-8 ($P = 0.026$). The result of sensitivity analysis is shown in Supplementary Table 4. Each dot represents a single case. $n = 51$ pairs (baseline and 4 W). **d** Line plots showing serial changes in serum concentrations of IL-8-high and Ang-2-high subgroups stratified by the immune subtype of tumor. Threshold, median at baseline. $^*p < 0.05$; $^{**}p < 0.01$; $^{***}P < 0.001$. ns not significant, PFS progression-free survival.

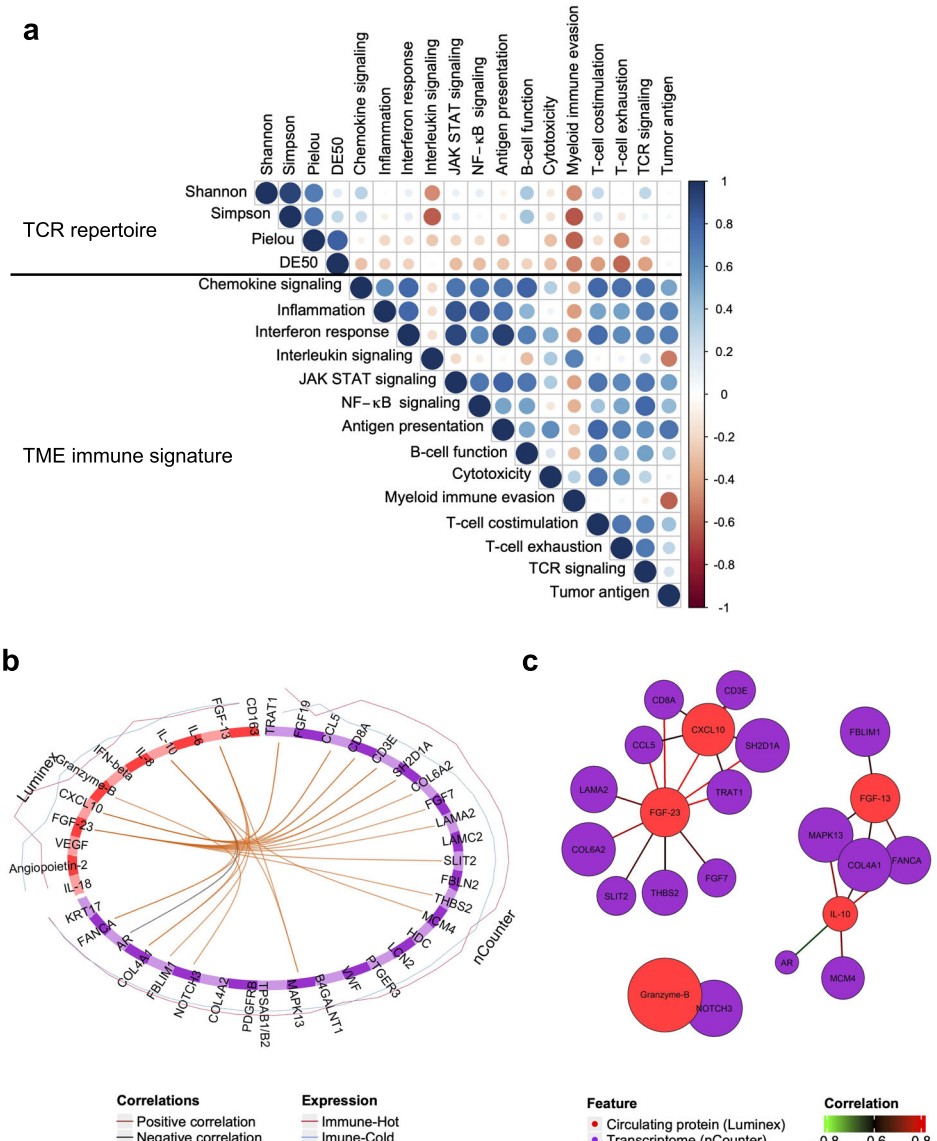

**Fig. 6 Integrative analysis of TCR repertoire, TME transcriptome and circulating protein in blood. a** Correlation matrix plot showing coefficients of diversity indexes of TCR repertoire and immune-related signature represented by ssGSEA score. $n = 12$. **b** Circos plot showing the strong positive and negative correlation between features of circulating proteins (Luminex) and features of TME transcriptome (nCounter) as indicated by the brown and black links. **c** Network plot highlighting three clusters including key features from each omics analysis. $n = 39$ (**b**, **c**). TCR T-cell receptor, TME tumor microenvironment, Shannon Shannon-Weaver index H', Simpson Inv. Simpson's index $1/\lambda$, Pielou Pielou's evenness, DE50 Diversity Evenness score.

a xenograft model of renal cell carcinoma[53], no other studies have followed serum levels of IL-8 during targeted therapy in clinical samples. On the other hand, IL-8 has increasingly been investigated as a predictive biomarker for immune checkpoint blockade, and early decreases in serum IL-8 are strongly associated with favorable outcomes[54]. Because both cancer cells and immunoinhibitory immune cells can represent a source of supply of IL-8 in the TME, a remaining issue would be to determine which of these factors affected by lenvatinib contributes more to the decrease in IL-8. In contrast with IL-8, angiopoietin-2 has been less investigated in the context of a drug target that enhances the efficacy of immunotherapy. Although the mechanisms remain unclear, a phase 3 trial has also found that angiopoietin-2 levels were reduced from baseline only in the lenvatinib-treated group ($n = 266$) compared to the sorafenib-treated group ($n = 127$)[55]. The leading hypothesis explaining the suppression of angiopoietin-2 production is the inhibition of tumor vascular endothelial function by dual inhibitory activity of VEGFR and

FGFR blockade[56]. FGF-FGFR signaling plays an important role in both tumor angiogenesis and antitumor immunity and may also be involved in resistance to VEGF inhibition[57]. Dual blockade of VEGF and FGF signaling by lenvatinib might, therefore, be advantageous as a potentiator of immunotherapy.

This study has several limitations. Although sample collection was performed prospectively and the amounts of collected biopsied tissues were sufficient for most of the planned analyses, the amount of RNA for TCR repertoire analysis was insufficient. As a result, more than half of the samples did not meet the RNA quality criteria of the experiment. Our preliminary results of increased T-cell diversity during therapy are quite promising[58], and validation in other cohorts with sufficient RNA (perhaps derived from surgically resected specimens) is mandatory. The incompleteness of multiplex imaging mass cytometry also hampered a comprehensive understanding of the TME. Rare earth-labelled antibodies including PD-L1 against tumor and immune cells or CD11c against dendritic cells did not work in the present

experimental setting. Alternative immunohistochemical staining was not possible because of the lack of remaining specimens. However, the role of existing biomarkers, such as PD-1 and PD-L1, is limited in the field of liver cancer[47], and the focus on other antigens has provided impetus for the proposal of new hypotheses. The final limitation is the absence of tissues other than liver and liver tumor. To perform a qualified study with homogeneous samples, patients lacking technically available lesions for liver tumor biopsy were excluded. Therefore, whether a similar approach will unravel immunomodulation in tumor tissues located in metastases to other organs remains unclear. However, the fact that the liver, where immune evasion and excessive immunosuppression are frequently established, showed immune activation by therapeutics represents a significant step toward the development of future therapies for HCC.

## Data availability

Annotations for the nCounter Tumor Signaling 360 Panel, which is used for the RNA expression analysis in the study, are contained in Supplementary Data 1. The Cancer Genome Atlas-tumor microenvironment classification platform are defined in detail in Supplementary Data 2. The datasets generated during the current study are available in the Figshare repository: https://doi.org/10.6084/m9.figshare.22215760.v3[59]. The main source data for the figures in this manuscript are included in Supplementary Data 3. All other data supporting the results of this study are available from the corresponding author upon reasonable request.

## Code availability

Custom codes, which are all written in R language and used in the current study, are available in the Figshare repository: https://doi.org/10.6084/m9.figshare.22215760.v3[59].

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

## Acknowledgements
This study was supported by Eisai Co., Ltd. We are grateful to the patients and their families who made this study possible. We wish to thank A. Fujimoto and K. Muramoto, former advisors at the start of this study who belong to Eisai Co., Ltd. We also wish to thank Y. Kato, a latest advisor of this study who belong to Eisai Co., Ltd. We gratefully acknowledge the assistance with sample preparation by the medical secretaries at Hiroshima University Hospital (H.S., N.Y., Y.K., A.O., and E.N.).

## Author contributions
H.A, K.C., and M.Y. conceived the study. M.Y. designed and directed the study, analyzed the data, and wrote the manuscript. A.O. designed experiments and analyzed data. C.N.H and H.N. contributed with bioinformatics for the study. K.A., Y.F., Y.T., S.U., H.F., T.N., E.M., W. Okamoto, D.M, T. K., M.T., and M.I. contributed by collecting specimens and clinical data. W. Ohishi and T. Kishi supported and performed serum analyses. M.K. and N.S. provided important suggestions and improved the manuscript. K.A. prepared pathological specimens and provided pathologic interpretations. S.O. supervised the whole project and provided advice. All authors discussed the results and reviewed the manuscript.

## Competing interests
The authors declare the following competing interests. K.C. received lecture fees from Bristol-Myers Squibb, MSD, AbbVie, Gilead, and Sumitomo Pharma and received grants from Bristol-Myers Squibb, Sumitomo Pharma, MSD, AbbVie, TORAY, Eisai, Janssen Pharmaceutical, Daiichi-Sankyo, Roche, Otsuka Pharmaceutical, and Mitsubishi-Tanabe Pharma during the conduct of the study. M.K. and N.S. are employees of Eisai Co., Ltd. All other authors declare no competing interests.

## Additional information

[1]Department of Gastroenterology, Graduate School of Biomedical and Health Sciences, Hiroshima University, Hiroshima, Japan. [2]Department of Clinical and Molecular Genetics, Hiroshima University, Hiroshima, Japan. [3]Department of Clinical Studies, Radiation Effects Research Foundation, Hiroshima, Japan. [4]Oncology Department, HQs, Eisai Co., Ltd, Tokyo, Japan. [5]Department of Anatomical Pathology, Hiroshima University Hospital, Hiroshima, Japan. [6]Department of Gastroenterology, Hiroshima Prefectural Hospital, Hiroshima, Japan. [7]Collaborative Research Laboratory of Medical Innovation, Hiroshima University, Hiroshima, Japan. [8]Hiroshima Institute of Life Sciences, Hiroshima, Japan. [9]RIKEN Center for Integrative Medical Sciences, Yokohama, Japan. ✉email: myamauchi@hiroshima-u.ac.jp; chayama@mba.ocn.ne.jp; oka4683@hiroshima-u.ac.jp

