## [Peer Review File · Communications Medicine]

Reviewers' comments:

Reviewer #1 (Remarks to the Author):

This article presented a multi-omics analysis of tumor and blood samples from 51 patients with advanced hepatocellular carcinoma during lenvatinib monotherapy and analyzed the corresponding serum circulating proteins, imaging mass cytometry and T-cell receptor (TCR) repertoire to validate the results. However, there are some problems, which must be solved before it is considered for publication.

1. The number of paired tumor biopsy and peripheral blood samples is unclear.
2. Transcriptome-based genomic alternation of lenvatinib and T-cell receptor repertoire in the HCC tumor have been reported respectively. The results of multi-omics analysis are not integrated. Please indicate any innovations that differ from previous studies.
3. The authors need to define the two TME immune subtypes. Interpretation should be deepened regarding the two immune subtypes of the TME (“Immune-Hot” and “Immune-Cold”). How does this differ from the hot and cold tumor staging reported in previous studies? The two immune subtypes lacked enough clinical sample validation.
4. The expression of whole 21 serum proteins is ambiguous and absent. The authors need to explain why CXCL10 is chosen from 21 serum proteins.
5. The results of the combined analysis of proteomics and TCR sequencing data were missing in the multi-omics analysis.

Reviewer #2 (Remarks to the Author):

In this study Yamauchi and colleagues perform an in-depth characterisation of human clinical samples from patients treated with Lenvatinib monotherapy. They provide compelling evidence that the pre-treatment TME subtype does not play a significant role in determining response to Lenvatinib and evidence for changes in the TME and circulation that correlate with response. Nevertheless, I have some issues for the authors to address prior to publication.

1. This is an interesting study however the narrative is confusing to follow, jumping between serum and tissue at different treatment stages. The manuscript would benefit from a simpler narrative grouping like data together. For instance Fig 2d analysis of serum IL8 and angio2 being able to stratify patients with poor prognosis is interesting however I fail to see how it links with hot or cold tumours survival curves identified in Fig 2c. It may be more suited after Fig 5b?
2. This is a highly descriptive manuscript relying heavily on bioinformatics analysis of bulk transcriptomics data. Further analysis of the imaging mass cytometry data or additional immunohistochemistry confirming the presence and/or perturbation of each immune cell classified by sequencing deconvolution analysis would be beneficial.
3. Fig 3 – is an interesting analysis of matched biopsy samples however they are relatively low in number n=3 hot vs n=5 cold. The authors need to provide further information on these samples including disease aetiology, TCGA-subtype, patient sex, age etc as these may be confounders that need to be taken into consideration
4. Fig 4g – TCR diversity data needs to be displayed in a different format as it is not clear what it is trying to show. Authors should also check text size in other figures as it is not always easy to read.
5. Figure 5 –authors should stratify these patients as either responders or non-responders. This may provide further insight into circulating biomarkers of response.
6. Sup 6a – can the authors please comment on why these regions of interest were chosen.
7. Authors should refrain from using 'CAFs' as this typically means cancer associated fibroblast
8. Imaging mass cytometry antibody table – antibodies are highlighted in the table which are not included as data in the manuscript. The authors should consider including data showing expression of immune checkpoints such as PD-1 and PD-L1.

Reviewer #3 (Remarks to the Author):

This paper studies lenvatinib monotherapy for hepatocellular carcinoma with imaging, TCR and RNA analyses. The main finding is clustering of the TME into two subtypes, which separate nicely. Pre- and post-treatment analyses show multiple immunological and signaling effects at high significances. The results and discussions are well supported by evidence, and the statistical data analysis is professionally performed and is valid. The paper is well written and results are comprehensively illustrated. The work is solid throughout.

My only minor comment is that in fig5 the differential analyses are perhaps too eager to jump to conclusions. I agree that the hypothesis testing does show significant differences, however the distributions might have outlier effects that might not be taken into account. Both Angiopoietin-2 and IL-8 decreases seem to stem from just one or two points, which could be outliers (see also CXCL10). Here sensitivity analysis could be useful to evaluate how robust the results are. I found no issues in other parts of the paper wrt this.

Detailed responses to reviewers' comments

Dear Editor and Reviewers,

We appreciate the helpful advice, which has greatly helped us improve the quality of our manuscript (COMMSMED-23-00128-A). The manuscript has been carefully revised according to the reviewers' comments. We believe that we have clearly addressed all the concerns, and we hope that you will find the paper suitable for publication in *Communications Medicine*.

Reviewer #1 (Remarks to the Author):

This article presented a multi-omics analysis of tumor and blood samples from 51 patients with advanced hepatocellular carcinoma during lenvatinib monotherapy and analyzed the corresponding serum circulating proteins, imaging mass cytometry and Tcell receptor (TCR) repertoire to validate the results. However, there are some problems, which must be solved before it is considered for publication.

1. The number of paired tumor biopsies and peripheral blood samples is unclear.

Author's response

Thank you for your suggestions. As the reviewer pointed out, it is difficult to determine the number of paired samples from the information in **Supplementary Fig. 1a**. We added the following text specifying the number of paired tumor biopsies and peripheral blood samples at each time point.

“Tissue samples were available at baseline in all but one case (n=50), while tissue samples during treatment were available for paired analysis at baseline and 4 weeks in 8 cases, and at baseline and post-progression in 12 cases. Blood samples (n=51) were all comparable with no missing data at any time point (page 6, lines 8-11)”.

2. Transcriptome-based genomic alternation of lenvatinib and T-cell receptor repertoire in the HCC tumor have been reported respectively. The results of multi-omics analysis are not integrated. Please indicate any innovations that differ from previous studies.

Author's response

Thank you very much for your observation concerning our multi-omics analysis. Please let us answer here in conjunction with our answer to Point 8. To summarize, the integrated analysis succeeded in visualizing what was difficult to see through single-omics analysis. Despite the small amount of repertoire data available and its diversity based on individual differences, we have managed to interpret the relationship of the TCR repertoire and the TME status. Specifically, we used the diversity indexes of each sample instead of sequencing reads and performed correlation analysis with tissue immune signatures represented by ssGSEA score (n=12). As shown in the newly added **Fig. 6a**, the variability of TCR repertoires in the TME (Shannon, Simpson and Pielou) is negatively correlated with the extent of immune evasion caused by myeloid cells.

This might signal that it is more important to control cancer-immune inhibition through myeloid cells such as M2-macrophages and myeloid-derived suppressor cells (MDSC) than through regulatory lymphocytes in the TME to enhance the efficacy of immunotherapy. Interestingly, clonality of the repertoire (DE50) also appears to be negatively associated with the T-cell exhaustion signature, including *PD-1(PDCD1)*, *LAG3* and *EOMES* genes.

Fig.6

To emphasize the significance of conducting data integration, we next combined tissue RNA expression and circulating immune proteins in peripheral blood, for which we had a large number of paired samples (39 pairs). By using a multivariate regression method, mixOmics framework^{#1}, we reduced the dimensionality of the data and explored key variables relating to the immune subtype. As shown in **Fig. 6b, c**, the analysis successfully extracted not only expected factors (e.g., serum CXCL10 and Granzyme B) to discriminate the immune subtype, but unexpected factors such as FGF23. Network analysis shows that serum FGF23, a ligand for FGFR4 (molecular target of lenvatinib therapy), may be closely related to serum CXCL10 via tissue chemokine (CCL5) and T-cell-associated molecule (TRAT1). Although the mechanism is not yet known, combining approaches in this way can provide deeper insight into the tumor microenvironment.

#1 Lê Cao, K.A., González, I. & Déjean S. integrOmics: an R package to unravel relationships between two omics datasets. *Bioinformatics* **25**, 2855-6 (2009).

#2 Rohart, F., Gautier, B., Singh, A. & Lê Cao, K.A. mixOmics: An R package for 'omics feature selection and multiple data integration. *PLoS Comput Biol* **13** (2017).

Fig. 6

We also added graphics that show sample scatterplots with reasonable correlation coefficients, the contribution of each factor to the first component, and a clustered image-map of the multi-omics signature in this data integration analysis (**Supplementary Fig. 7c-e**).

Supplementary Fig. 7

To summarize the above results, the following text was inserted at the end of the Results section (pages 15-16, lines 14-6). Explanations were added to corresponding figure legends.

“Multi-omics integration expands the possibility for deeper understanding of the TME: We finally confirmed the relationship between single-omics by several data coupling approaches. Correlation analysis comparing the TCR repertoire and the TME transcriptome at baseline showed a negative relationship between diversity indexes (Shannon, Simpson and Pielou) of repertoire and immune evasion signature by myeloid cells (**Fig. 6a**). It also showed a negative correlation between repertoire clonality (DE50) and T-cell exhaustion signature including PD-1, LAG3, and EOMES genes. The large

number of paired samples when comparing immuno-transcriptome and circulating proteins in blood allowed for data integration based on multivariate regression 33,34. The analysis successfully extracted expected factors (e.g., serum CXCL10 and Granzyme B) to discriminate the immune subtype, as well as unexpected factors such as FGF23 (**Fig. 6b, Supplementary Fig. 7c-e**). Network analysis showed that serum FGF23, a ligand for FGFR4, may be closely related to serum CXCL10 via chemokine (CCL5) and T-cell-associated molecule (TRAT1) expression in the microenvironment (**Fig. 6c**)”.

We also modified the following text in the Discussion section.

“Our integrative multi-omics approach and sequential observations in different immune subtypes provide putative mechanistic insights for better understanding of the nature of the TME and for developing new strategies enhancing the efficacy of immunotherapy (page 16, lines 14-17)”.

3. The authors need to define the two TME immune subtypes. Interpretation should be deepened regarding the two immune subtypes of the TME (“Immune-Hot” and “Immune-Cold”). How does this differ from the hot and cold tumor staging reported in previous studies? The two immune subtypes lacked enough clinical sample validation.

Author’s response

Thank you very much for your valuable advice. We have cross-referenced the two immune subtypes (Immune-Hot and Cold) with previously published immune-related scores and taxonomies that have established ratings. In the new **Fig. 1c**, we have added the immune score (Yoshihara K, 2013^{#3}) annotations across the top of the heatmap. This clearly indicates that the result of our clustering method correlates with the abundance of immune cells in the TME.

Fig. 1c

Additionally, in the new **Supplementary Fig. 1c**, we have inserted the Immune-specific class (Sia D, 2017^{#4}) and the Molecular subclass (Hoshida Y^{#5}) calculated by the NTP platform^{#6} as top annotations. By incorporating information from well-established

classification schemes, we intended to validate our clustering method initially using the TCGA-LIHC dataset. This inclusion supports the successful validation of our method.

Supplementary Fig. 1c

In summary, the Immune-Hot subtype includes most of the Immune-specific class by Sia and the HCC-subtype-S1 (aberrant activation of the WNT signaling pathway) by Hoshida. Conversely, the Immune-Cold subtype appears to correlate with Hoshida-subtype-S2 (more aggressive, with Myc and AKT activation), which is consistent with the context described in the following section (around **Fig. 2a**). Although the reviewer pointed out the need for clinical sample validation in the field of translational research, our primary objective in this TME clustering is to elucidate the behavior of immune cells during treatment in different TME states through subsequent analysis of RNA and protein expression. Unfortunately, we do not possess sufficient new sample series for additional experiments. We hope that the reviewer will consider the analysis of the TCGA dataset as a valid confirmation.

- #3 Yoshihara, K., et al. Inferring tumour purity and stromal and immune cell admixture from expression data. *Nat Commun* **4**, 2612 (2013).
- #4 Sia, D., et al. Identification of an Immune-specific Class of Hepatocellular Carcinoma, Based on Molecular Features. *Gastroenterology* **153**, 812-826 (2017).
- #5 Hoshida, Y., et al. Integrative transcriptome analysis reveals common molecular subclasses of human hepatocellular carcinoma. *Cancer Res* **69**, 7385-92 (2009).
- #6 Hoshida, Y. Nearest template prediction: a single-sample-based flexible class prediction with confidence assessment. *PLoS One* **5**, e15543 (2010).

We added the following text to the Result section and modified the Method section and the Figure legend section, accordingly.

“Another established score for quantifying immune cells in the TME, Immune score, also showed a strong correlation with our clustering results (upper column, **Fig. 1c**). To

further validate the reliability of our panel-based assay, which had limited mounted RNA, we cross-referenced a large dataset of RNA-seq from the TCGA-LIHC cohort using the same approach (**Supplementary Fig. 1c**). The classifications made by the nCounter RNA set were generally consistent with the results obtained using the TCGA-TME RNA set, with the exception of a small population within the TCGA-TME-Fibrotic subtype, in which the signatures of anti-tumor immune infiltrates were relatively overexpressed. Furthermore, the Immune-Hot subtype predominantly included cases classified as Immune-specific class by Sia and HCC-subtype-S1 by Hoshida. In contrast to the Immune-Hot subtype, the Immune-Cold subtype appeared to correlate with Hoshida-subtype-S2, which is a more aggressive subtype characterized by activating proliferation pathways (page 7, lines 10-22)”.

4. The expression of whole 21 serum proteins is ambiguous and absent. The authors need to explain why CXCL10 is chosen from 21 serum proteins.

Author’s response

Firstly, we would like to apologize for a simple error regarding the number of Luminex analytes. Because the actual number of whole serum proteins is 25, we have corrected the Method section and the Result section accordingly. As the reviewer suggested, the validity of extracting CXCL10 as a biomarker is very important. We added new figures showing the expression of whole 25 serum proteins in **Fig. 5a** and **Supplementary Fig. 6a**, as shown below. We also added corresponding explanations to the main text and figure legends.

“In the context of optimization in systemic therapy for HCC, we returned to whole-patient analysis and explored serum assays for circulating chemokine/angiogenic factors. Of 25 serum proteins quantified by the Luminex system, the CXCL10 level at baseline allowed the differentiation of two immune subtypes (**Fig. 5a, Supplementary Fig. 6a**). Although sensitivity analysis showed that the statistics might be affected by possible outliers, the validity of serum CXCL10 as a biomarker shows promise subject to validation in a large number of cases or in combination with other chemokines for the following reasons. As a key modulator for chemoattraction of immune cells, promotion of T-cell function, and antitumor activity in the local TME²⁵, CXCL10 may offer a surrogate for presuming the tumor immune subtype (page 13, lines 10-19)”.

Fig. 5a

Supplementary Fig. 6a

5. The results of the combined analysis of proteomics and TCR sequencing data were missing in the multi-omics analysis.

Author's response

Thank you for your suggestion. As mentioned in our response to your Point 2, we have added our data integration result for at the end of the Result section.

Reviewer #2 (Remarks to the Author):

In this study Yamauchi and colleagues perform an in-depth characterisation of human clinical samples from patients treated with Lenvatinib monotherapy. They provide compelling evidence that the pre-treatment TME subtype does not play a significant role in determining response to Lenvatinib and evidence for changes in the TME and circulation that correlate with response. Nevertheless, I have some issues for the authors to address prior to publication.

1. This is an interesting study however the narrative is confusing to follow, jumping between serum and tissue at different treatment stages. The manuscript would benefit from a simpler narrative grouping like data together. For instance Fig 2d analysis of serum IL8 and angio2 being able to stratify patients with poor prognosis is interesting however I fail to see how it links with hot or cold tumours survival curves identified in Fig 2c. It may be more suited after Fig 5b?

Author's response

Thank you for your comment about readability with regard to tissue and blood analysis. As you pointed out, we have moved **Fig. 1d** (bar plot, CXCL-10) and **Fig. 2d** (survival curves, IL-8 & Ang-2) in the original manuscript to revised **Fig. 5**, because we thought it would help readers' understanding to summarize the blood analysis in the latter half. As mentioned in the response to Reviewer #1, distributions of all Luminex analytes are added to these revised violin plots (revised **Fig. 5a**, revised **Supplementary Fig. 6a**).

Fig. 5

We also moved contents in **Supplementary Fig. 3** in the original manuscript explaining the result of blood analysis to revised **Supplementary Fig. 6**. Accordingly, the text explaining those figures was moved to the appropriate location.

Supplementary Fig. 6

2. This is a highly descriptive manuscript relying heavily on bioinformatics analysis of bulk transcriptomics data. Further analysis of the imaging mass cytometry data or

additional immunohistochemistry confirming the presence and/or perturbation of each immune cell classified by sequencing deconvolution analysis would be beneficial.

Author’s response

Thank you for your valuable advice. We would like to respond to your comments here, along with our answer to your Point 8, which will come later. In the original manuscript, we had reduced most of the imaging mass cytometry data to simplify our argument. However, as the reviewer mentioned, we are now convinced that presenting the overall distribution of IMC expression and the characteristic images at baseline would enhance the validity of the study. First, the expression of all antibodies has been displayed separately for the tumor, adjacent non-tumor, and stromal areas in **Supplementary Fig. 2b, c**.

Supplementary Fig. 2

Next, we added the IMC results to **Fig. 2** to show that our TME clustering method is valid not only at the transcriptome level, but at the protein level (**Fig. 2d**).

Fig. 2d

In addition, following sentences were added to explain these figures and mention that some degree of immune cell infiltration can be seen in the peri-tumoral and stromal areas even in Immune-Cold cases.

“We also checked the phenotype of two immune subtypes in terms of protein-level expression. A list of corresponding clones and conjugated metal reporters used in the experiments is shown in **Supplementary Table 4**. As shown in **Fig.2d** and **Supplementary Fig. b, c**, Immune-Cold subtype tumor has low level expression of immune-related protein in the TME compared to Immune-Hot subtype. However, some cases have modest to high levels of activation of immune infiltrates especially in adjacent non-tumor liver tissue and stroma accompanied by activated VEGF and FGF signaling (pages 8-9, lines 20-5)”.

Finally, we have added representative images regarding beta-catenin as **Fig. 4e**, which might suggest immune modification other than changes observed in Tregs and M2-type macrophages.

Fig.4

We also added the following sentences to explain the limitation.

“Generalization across multiple cases could not be confirmed because of the limited number of ROIs. We also had an interesting observation suggesting suppression of beta-catenin signaling and activation of the TME (**Fig. 5e**) (page 12 lines 13-15)”.

3. Fig 3 – is an interesting analysis of matched biopsy samples however they are relatively low in number n=3 hot vs n=5 cold. The authors need to provide further information on these samples including disease aetiology, TCGA-subtype, patient sex, age etc as these may be confounders that need to be taken into consideration.

Author’s response

We appreciate the comments concerning these critical points in the sub-analysis with a small number of cases. As suggested by the reviewer, we have added a new figure (revised **Fig. 3a**) which includes detailed patient information. We believe that this will help the reader recognize that although the number of paired cases analyzed was limited, they are well representative of the population. We have added a description of the median and range of age and the median progression-free survival to the figure legend.

Fig. 3a

4. Fig 4g – TCR diversity data needs to be displayed in a different format as it is not clear what it is trying to show. Authors should also check text size in other figures as it is not always easy to read.

Author’s response

Thank you for your recommendation. We have added the following figures as revised **Fig. 4f** to convey our intent. We believe that these figures would help readers intuitively grasp the changes in repertoire diversity and clonality during treatment.

Fig. 4

We modified the following text accordingly.

“Although the number of paired samples available for analysis was limited, in two of three pairs, three diversity indexes, which are widely used for comparing diversity between samples, were increased at 4W (**Fig. 4f**) (page 12, lines 17-19)”.

We also added a new explanation specifying the increase in the clonality of the repertoire: “Furthermore, an increase in the clonality of the repertoire, a possible tumor-specific response, was also observed in on-treatment samples (page 12, lines 19-20)”.

As the reviewer noted, there were many small fonts in the original figures; we have corrected them for readability as shown in **Fig. 4g** and **Supplementary Fig. 5b**. In addition, most of the other figures in the manuscript have also been modified to make them as easy to read as possible by adjusting their placement and fonts.

Fig. 4

Supplementary Fig. 5

5. *Figure 5 –authors should stratify these patients as either responders or non-responders. This may provide further insight into circulating biomarkers of response.*

Author’s response

Thank you for your valuable advice. As the reviewer suggested, we have added figures illustrating the variation of circulating proteins during treatment stratified by responsiveness to lenvatinib (**Supplementary Fig. 7**). We also added the following sentences to explain our interpretation in the Result and Discussion sections.

“The trends of increase and decrease in circulating proteins are similarly observed in both the Responder and the Non-Responder group (page 15, lines 10-12)”.

“In addition, the trends of changes in circulating proteins regardless of tumor response suggest that the immune modification induced by lenvatinib is not due to its antitumor effect, but rather is evoked by inhibition of signaling pathways (page 18, lines 8-11)”.

Supplementary Fig. 7a

6. *Sup 6a – can the authors please comment on my these regions of interest were chosen.*

Author's response

Thank you for drawing our attention to the definition of regions of interest (ROIs). Actually, ROIs in the Hyperion analysis were randomly selected and laser-ablated by a technician independent of data processing, referring to the HE images and carefully avoiding necrotic areas. Among those ROIs, **Supplementary Fig. 6a** is a representative ROI showing the infiltration of CD8-positive cells into the viable tumor during treatment, even in cases that were evaluated as Immune-Cold at baseline (FL40). We have added explanations to the Figure legend section in order to better convey our intent. The channel-specific display shows that each antibody is working properly. To avoid arbitrary ROI selection, 2-4 ROIs were selected from each case and the counts of antibody-positive cells are shown as boxplots. As mentioned in the response to Comment 1, the order in which some figures are presented has been changed, so **Supplementary Figure 6a** is presented as **Supplementary Figure 5a** in the revised version.

7. *Authors should refrain from using 'CAFs' as this typically means cancer associated fibroblast*

Author's response

Thank you for your important advice. As the reviewer pointed out, all "CAFs" in the manuscript have been fully spelled out as "circulating chemokine/angiogenic factors" or replaced with "serum proteins" and "circulating proteins" to avoid confusion.

8. *Imaging mass cytometry antibody table – antibodies are highlighted in the table which are not included as data in the manuscript. The authors should consider including data showing expression of immune checkpoints such as PD-1 and PD-L1.*

Author's response

Thank you for your request. As mentioned in our response to your Point 2, we have included all IMC antibodies in the new figure and analyzed expression of each protein for the major TME phenotypes.

Reviewer #3 (Remarks to the Author):

This paper studies lenvatinib monotherapy for hepatocellular carcinoma with imaging, TCR and RNA analyses. The main finding is clustering of the TME into two subtypes, which separate nicely. Pre- and post-treatment analyses show multiple immunological and signaling effects at high significances. The results and discussions are well supported by evidence, and the statistical data analysis is professionally performed and is valid. The paper is well written and results are comprehensively illustrated. The work is solid throughout.

My only minor comment is that in fig5 the differential analyses are perhaps too eager to jump to conclusions. I agree that the hypothesis testing does show significant differences, however the distributions might have outlier effects that might not be taken into account. Both Angiopoietin-2 and IL-8 decreases seem to stem from just one or two points, which could be outliers (see also CXCL10). Here sensitivity analysis could be useful to evaluate how robust the results are. I found no issues in other parts of the paper wrt this.

Author’s response

Thank you very much for your insightful remarks. As the reviewer advised, we performed sensitivity analysis on the Luminex analytes that may vary widely among individuals and might include possible outliers. The results were added as **Supplementary Table 5**. Because the comparisons are made by rank-based tests, the results seem robust even after excluding one or two potential outliers regarding changes in the levels of IL-8 and angiopoietin-2 between baseline and 4W.

Supplementary Table 5.

Circulating protein	Excluded value (ID, value at baseline)	P-value
Immune subtype (Fig. 5a)		
CXCL10	FL38, 340 pg/ml	0.060
Change during treatment (Fig. 5c)		
CXCL10	FL38, 340 pg/ml	<0.01
IL-8	FL4, 725 pg/ml	0.025
IL-8	FL4 and FL45, 387 pg/ml	0.063
Angiopoietin-2	FL48, 12197 pg/ml	0.008
Angiopoietin-2	FL48 and FL-46, 9118 pg/ml	0.013

On the other hand, as the concern about usefulness of the level of CXCL10 at baseline remained, we added the following sentences.

“Although sensitivity analysis showed that the statistics might be affected by possible outliers, the validity of serum CXCL10 as a potential biomarker is worth validation in a large number of cases or in combination with other chemokines for the following reasons (page 13, lines 14 to 16)”.

REVIEWERS' COMMENTS:

Reviewer #1 (Remarks to the Author):

The authors have carefully revised the manuscript based on the reviewers' suggestions. The manuscript is now much improved and suitable for publication in Communications Medicine.

Reviewer #2 (Remarks to the Author):

The authors have sufficiently addressed my concerns and I am pleased to recommend it for publication.

Reviewer #3 (Remarks to the Author):

The authors have addressed my concerns. I have no further issues with the paper.